# Early high-sensitivity troponin elevation in predicting short-term mortality in sepsis: A protocol for a systematic review with meta-analysis

Santiago Ferrière-Steinert[1], Joaquín Valenzuela Jiménez[1], Sebastián Heskia Araya[1], Thomas Kouyoumdjian[1], José Ramos-Rojas[2], Abraham I. J. Gajardo[3,4] *

1 Faculty of medicine, School of medicine, Universidad de Chile, Santiago, Chile, 2 Faculty of Medicine, Dentistry School, Clínica Alemana-Universidad del Desarrollo, Santiago, Chile, 3 Department of Internal Medicine, Intensive Care Unit, Hospital Clínico Universidad de Chile, Santiago Chile, 4 Faculty of Medicine, Institute of Biomedical Science, Program of Pathophysiology, Universidad de Chile, Santiago, Chile

* aij.gajardo@gmail.com

**Editor:** Sonu Bhaskar, Global Health Neurology Lab / NSW Brain Clot Bank, NSW Health Pathology / Liverpool Hospital and South West Sydney Local Health District / Neurovascular Imaging Lab, Clinical Sciences Stream, Ingham Institute, AUSTRALIA

## Abstract

### Background

Sepsis is a common admission diagnosis in the intensive care unit (ICU). The Sepsis-3 consensus associates sepsis diagnosis with acute organ dysfunction. In these patients troponin elevation is a well-established phenomenon, but its clinical significance is not settled, as no systematic review has addressed the prognostic significance of the increasingly prevalent high-sensitivity troponin assays in acute organ dysfunction setting.

This study aims to clarify the association between early serum troponin levels in high-sensitivity assays with short-term mortality risk in septic patients with acute organ dysfunction.

### Methods

We will systematically search PubMed, Scopus and Embase for original articles; additionally, a manual search will be carried out through relevant literature. Generally, studies will be deemed eligible for inclusion if they evaluate the association between high-sensitivity troponin in the first 24 hours of admission and ICU, 30-days, or In-hospital mortality; in patients with septic shock or sepsis related to acute organ dysfunction. Two reviewers will independently select studies and extract the data. A meta-analysis for mortality outcome will be performed for comparative data regarding two effect measures: Odd ratios and Standardized Mean differences.

### Discussion

This study will provide further evidence about the role of high-sensitivity troponin assays in predicting mortality in septic patients; potentially helping to guide further research and yielding valuable information for patient assessment.

**Data Availability Statement:** No datasets were generated or analysed during the current study. All relevant data from this study will be made available upon study completion.

**Funding:** This study was funded by the Government of Chile, through the grant: Initiation in research FONDECYT grant 11241548, National Agency of Research and Development (https://anid.cl/about-us/); Awarded to Abraham I. J. Gajardo. The funders had no role in study design, data collection and analysis, decision to publish, or preparation of the manuscript.

**Competing interests:** The authors have declared that no competing interests exist.

Conclusion about the certainty of evidence will be presented in a ´Summary of findings´ table.

## Trial registration

PROSPERO registration:

(CRD42024468883).

## 1. Background

Sepsis is the most common mortality cause in Intensive Care Unit (ICU) patients in the United States. A fourth of patients diagnosed with sepsis will die within their hospitalization, rising to a half within the septic shock group [1]. Furthermore, sepsis-related deaths have been on the rise in the US and worldwide [1–3], and are expected to continue to do so.

Sepsis definitions have varied throughout the years. Sepsis-3 consensus, agreed in 2016, defines sepsis as a life-threatening organ dysfunction (assessed by SOFA score) in behalf of an infection; disposing of the former category of *severe sepsis* while preserving *septic shock* [4]. Thus, compared with previous sepsis definitions, patients diagnosed with sepsis-3 criteria are exposed to an increased mortality and adverse events risk [5].

Sepsis is associated with complications and dysfunction of several systems [6]. At the cardiac level, sepsis is well-documented to be associated with acute myocardial injury independent of coronary perfusion abnormality [7]. This phenomenon is primarily attributed to inflammatory and cardio-depressant factors that act indirectly and directly on the cardiomyocyte [7,8]. Furthermore, when the myocardial injury ensues in diastolic and/or systolic dysfunction, it is considered Septic Cardiomyopathy [7]. Although there is no consensus on the criteria for this entity, commonly accepted cardinal features are: The absence of an underlying coronary etiology, the acute presentation, and the reversibility of the dysfunction [7–9].

Cardiac troponins are widely used in the diagnosis of acute coronary syndrome and other cardiac pathologies presenting with myocardial injury [10], including sepsis. High-sensitivity cardiac troponin assays (hs-cTn) are usually defined by their capacity to detect troponin in at least 50% of healthy individuals [10,11]. In recent years hs-cTn have increasingly replaced conventional assays since they have both a lower limit of detection, and a lower and categorically defined normality cut-off. Thus, a hs-cTn level above the 99th percentile for the healthy population allows for a precise differentiation between *normal* and *elevated* groups [11].

Troponin elevation in septic patients is well-established but debated as a prognostic risk factor for clinical outcomes [7,12]. Three meta-analyses have been published on the issue [13–15], all showed a significant association of troponin elevation with increased mortality, but suffer from many problems of published data on troponin and mortality in sepsis: Lack of strict or similar criteria for sepsis diagnosis, heterogeneity of sampling time, usage of conventional rather than hs-cTn essays, and absence of controlling of confounders [7]. Moreover, the emergence and consolidation of Sepsis-3 definition, and the rise of hs-cTn assays could impact on the stratification of patients, modifying the prognostic association of the biomarkers with mortality risk [16]. Consequently, it is currently unknown if elevation in hs-cTn levels is associated with mortality risk in patients with sepsis by Sepsis-3 consensus.

This study aims to assess the association of early high-sensitivity serum troponin levels with short-term mortality in patients admitted to ICU because of sepsis with acute organ dysfunction.

## 2 Methods

### 2.1. Registration

The protocol was registered on the International Prospective Register of Systematic reviews (CRD42024468883). This protocol aligns with Preferred Reporting Items for Systematic review and Meta-Analysis Protocols (PRISMA-P) statement [17], and follows Riley´s guide for systematic review of prognostic factors [19] (see S1 Checklist). Deviations from protocol will be recognized in the final article.

### 2.2. Literature search

With the help of a knowledgeable librarian a search strategy was created: Pubmed, Scopus and Embase will be searched for primary articles published in any language and involving humans, with no date restrictions. The prepared search strings combine free text and MeSH/EMTREE terms for all words related to troponin and sepsis (see Table 1). No additional filters besides the ones described in the strings will be utilized.

Subsequently, we will manually scrutinize references of included studies and previous reviews on the topic to retrieve missing pertinent papers.

Searches will be re-run just before the final analyses to check for new studies that meet the inclusion criteria.

### 2.3. Study selection

In accordance to CHARMS guidance [18], the research question was formulated with the Population/Index prognostic factor/Comparator prognostic factors/Outcome/Timing/Setting (PICOTS) system, a modified version of the traditional PICO, where the *P* and *O* remains the same as the original, but the *I* stands for *Index prognostic factors* and *C* for other prognostic factors that can be considered as *Comparators* [19]. (See Table 2 for details on the defined scope of the systematic review). For the determination of inclusion and exclusion criteria Table 3 was constructed.

Based on this criteria, after deleting duplicates the corpus of studies will be scanned in Rayyan [20] by two independent reviewers, discrepancies will be solved by a third reviewer. Studies will be discarded based on 1 criterion only.

Initially, studies will undergo screening based on their title and abstract. Papers that meet the criteria during this preliminary assessment, or in cases where such analysis appears insufficient, will undergo further evaluation through full text assessment by the same reviewers for final selection.

Basically, interventional and observational primary studies will be deemed eligible if they reported some form of association between troponin levels and mortality in a general clinical context. Interventional studies will be handled as observational cohorts, extracting associations

**Table 1. Search strings utilized.**

| | |
|---|---|
| Pubmed string | (((("troponin i"[MeSH Terms]) OR ("troponin"[MeSH Terms])) OR ("troponin c"[MeSH Terms])) OR ("troponin t"[MeSH Terms]) OR ("troponin*"[All Fields]) AND ((humans[Filter]))) AND (((("sepsis"[MeSH Terms]) OR ("shock, septic"[MeSH Terms])) OR (Systemic Inflammatory Response Syndrome[MeSH Terms]) AND ((humans[Filter])))) |
| SCOPUS string | (TITLE-ABS-KEY (troponin) AND TITLE ((sepsis) OR (septic AND shock)) AND KEY ((sepsis) OR (septic AND shock))) |
| EMBASE string | 'sepsis'/exp/mj AND troponin* AND [humans]/lim |

**Table 2. PICOTS for the scope of the systematic review.**

|  | Definition |
|---|---|
| Population | Adults diagnosed with sepsis with acute organ dysfunction |
| Index prognostic factor | High-sensitivity troponin levels in serum |
| Comparator | Studies with or without comparators will be included. |
| Timing | Troponin sampling upon admission or within the first 24 hours. In-hospital or 30-days mortality |
| Outcome | All-cause mortality |
| Setting | Emergency department (ED) or Intensive Care Unit (UCI) |

for both arms when previous evidence suggests that the intervention is not directly confounding the relation between troponin and mortality; if such a confounding bias mechanism is deemed plausible, only the placebo arm data will be utilized. We will only include studies that report on patients diagnosed either with sepsis by Sepsis-3 criteria [4], with Severe sepsis by Sepsis-1 or Sepsis-2 [21,22], or Septic shock by any of them [4,21,22]. Selection will be restricted to studies that specify the utilization of any hs-cTn assay in the first 24 hours after hospital admission, or whose reported limit of detection we consider high-sensitivity (i.e. < 9 ng/L)[10]. In the case that the latter information about the assays is omitted, we will exclude studies if they fail to meet a normality cutoff of 20 ng/L or lower, since they will be regarded as

**Table 3. Inclusion and exclusion criteria for the systematic review.**

|  | Inclusion criteria | Exclusion criteria |
|---|---|---|
| **P**opulation | • Individuals >18 years old of any gender and any ethnicity in any country<br>• Individuals that were diagnosed with sepsis or septic shock according to sepsis-3 criteria, or with Severe sepsis or Septic shock according to Sepsis-1 or Sepsis-2 criteria. | • Studies with mixed or uniquely underaged samples<br>• Studies with ≤ 10 events (i.e. deaths) per study<br>• Sepsis is not the primary diagnosis.<br>• Letters to the Editor, Case reports, reviews, meta-analysis, and non-primary studies in general<br>• Studies restricted to a specific pathology (e.g studies focusing exclusively on COVID or Cancer patients)<br>• The paper refers to a sample already used by another paper in the corpus of studies |
| **I**ndex prognostic factor | • Studies explicitly declaring the utilization of any High-sensitivity cardiac troponin (cTnT, cTnI, or cTnC) essay<br>• If no explicit statement is made on the sensitivity, essays will be regarded as high-sensitivity if their declared Limit of Detection (LoD) is ≤ 5 ng/L | • Studies not publishing extractable numeric data on troponin measurements, who fail to answer attempts to be contacted.<br>• Studies explicitly declaring the utilization of non high-sensitivity essays<br>• If no statement is made on the LoD as well, studies with normal limit cutoff > 20 ng/L will be excluded.<br>• Studies that fail to publish either the sensitivity of their essays, the Limit of Detection (LoD) or their normal cutoff value; and fail to answer contact attempts |
| **C**omparator | Not applicable | Not applicable |
| **O**utcome | • All-cause mortality reported through hospital reports or other validated source | • Studies that didn´t report any form of association between the pertinent troponin levels and the pertinent mortality. |
| **T**iming | • First troponin measurement within the first 24 hours of admission<br>• Studies that include In-hospital mortality or follow-up mortality at most 30 days from admission. | • Studies where the first reported troponin levels measurement occurred beyond the first 24 hours after admission. |
| **S**etting | • Patients in ICU or Emergency department (ED) | • Other settings, including non-clinical context (e.g. experimentation) |
| Language | • Studies in any language | • No restriction because of language will be made |
| Publication status | • Published studies, conferences or abstracts. | • Other publication status, including retracted papers.<br>• Published abstracts where no access to the cohort characteristics could be obtained after appropriate contact attempts. |
| Species | • Human studies | • Animal studies<br>• In vitro studies |

**Table 4. Data extraction form.**

| Article and Author | Year | Number of patients | Setting and country | Eligibility and recruiting methods (multicenter or monocenter) | Age | Male (%) | Study type | Excluded comorbidities | Sepsis/Septic shock definition |
|---|---|---|---|---|---|---|---|---|---|
| Shock (%) | Mortality (%) | Mortality time follow-up | Effect measure(s) with Standard Error(s) | Troponin type used and normality cutoff | | | Type of statistical analysis | Variables in multivariate analysis (if any) | Quality Score (QUIPS) |

having a cutoff over the 99th percentile of the healthy population and therefore considered as non hs-cTn assay, based on European Society of Cardiology guidelines [10].

The study selection process will be reported using the PRISMA flowchart, providing a clear visual representation of the inclusion and exclusion criteria application [17].

## 2.4. Data extraction

Based on CHARMS-PF checklist [18] we designed a standardized data extraction form, containing the relevant characteristics of studies to be included (Table 4). Data will be extracted by two reviewers. Types of reported associations suitable to be extracted for this table are: 1) "dose-response": data reported as odds, risk, or hazard ratio per unit increase in exposure; 2) "category/quantile based": numbers (ideally 2x2 tables) or ratios comparing groups as defined by quantiles or categories of exposure; 3) "means": data reported as means or mean differences in exposure, comparing those presenting and not presenting the event [23]. If a study presents more than one of these associations, we will extract all of them.

We will collect extracted effect measures together with their standard errors (SE). If SEs are not explicitly available, we will derive them from metrics such SDs, exact p-values, or confidence intervals [24]. We will collect adjusted and unadjusted effect measures separately. If many adjusted models are presented, we will extract the one which fits better the core adjustment factors selected (see Risk of Bias section). If effect measures for several time points are presented, we will prioritize ICU and 28-mortality (in that order) over in-hospital mortality. If effect measures for several hs-cTn samples are presented, we will prioritize the sample extracted closest to ED/ICU admission. Data only presented as figures will be extracted with WebPlotDigitizer software [25].

Authors will be contacted to request critical (i.e. data mandatory for PICOTS assessment) missing or not reported data regarding associations of interest. In case of no response, the article will be excluded from this study.

We will present a summarized version of this table in the final review.

## 2.5. Risk of bias assessment

Bearing in mind the prognostic nature of the review and taking into account previously published meta-analysis [13–15], we can expect that mainly observational studies will be included.

Two authors will independently assess the risk of bias (RoB) applying the QUIPS (quality in prognostic factor studies) tool [26], and then discuss together their assessments in order to decide the final score. In QUIPS RoB assessments are made within a set of 6 domains: (1) Study Participation, (2) Study Attrition, (3) Prognostic Factor Measurement, (4) Outcome Measurement, (5) Study Confounding and (6) Statistical Analysis and Reporting. We defined Study participation, Prognostic Factor Measurement and Study Confounding as the key domains for our assessment, and we specified criteria for each of these domains. For the confounder domain RoB evaluation, we will look for 3 core adjustment factors in each study: severity of sepsis, age, and comorbidities (any cardiac or renal).

Classification for each RoB item could be 'low', 'moderate', or 'high'. Moreover, this assessment will include an overall judgment concerning the total RoB quality of each study, which will be visualized in the review.

## 2.6. Statistical analysis

We will investigate the mortality outcome with two effect measures: Mortality risk and Mean difference in troponin distribution.

1) Regarding mortality risk analysis: Whenever feasible a 2x2 contingency table will be gathered from available data, with one axis for survivor vs non-survivor groups, and other for elevated vs normal troponin level group, according to the cut-off defined by every study for each specific hs-cTn assay. If the sample is divided into more than 2 groups or categories (e.g. quantiles) it will be dichotomized around the closest match to a normality cutoff of 15 ng/L [27]. We will add 0.5 to each cell in any table that contains one or more zero values [24]. We will compute an Odds Ratio (OR) with its SE from this contingency table for each study. For studies not presenting sufficient data for a contingency table we will extract explicitly reported unadjusted ORs with their SEs. ORs from studies handling hs-cTn as a continuous variable (i.e. dose-response) will be scaled to an OR per 50 ng/L of increment. This magnitude was considered representative of the difference between the group usually referred to as "normal" compared to the "elevated" troponin level group–based on the reported troponin distributions of pertinent studies known to us and in accordance with our clinical expert [28–30].

Results from studies reporting Hazard Ratio (HR) or Relative Risk (RR) will be presented separately. Only if the event probability in the control group is <0.2, and the HR or RR are below 2.5, the effect measures will be interpreted as numerically equivalent to an OR and reported together with the other studies [31].

With these effect measures we will perform a structured quantitative analysis (meta-analysis) for the pooled unadjusted mortality effect.

ORs of adjusted (i.e. multivariate) regressions will be included in a separate model for the adjusted mortality effect regardless of the set of confounders controlled for. We will report variables incorporated in the regression for each study in a table.

2) For difference in troponin levels analysis: Studies presenting mean difference data in troponin level by survivor status will be meta-analyzed in a single model using Hedges´ g Standardized Mean Differences (SMD).

All models will be generated using R-4.3.1 package *metafor* [32], utilizing the random-effects model and the inverse variance weighting method [24].

Heterogeneity will be assessed and reported by I2 statistic, where >50% and a p-value < 0.10 are of concern and will be further explored [24].

Studies not presenting minimum data (e.g. missing SE) will be included in the narrative synthesis and compared to the results of the meta-analysis to the extent possible.

For the sake of replicability all formulas, raw data, code, calculations, model specifications and estimates shall be included either in the article itself or an online supplement.

## 2.7. Subgroup and sensitivity analysis

Models will undergo a comprehensive heterogeneity assessment. Since reviews of prognostic studies usually can result in high I2 regardless of similar point estimates [33], we will favor visual inspection over statistical criteria.

If in any of the model's heterogeneity is deemed substantial, we will run the following Subgroup analysis:

1. **Subgroups by shock status.** To assess whether effect measures and heterogeneity is significantly dependent on the degree of organic dysfunction, we will proceed with a subgroup analysis between studies restricting inclusion criteria to septic shock only, versus studies that encompass patients with sepsis, severe sepsis and septic shock.

2. **Subgroup by type of troponin.** Despite the fact that troponin essays have longed being considered equivalent in the acute coronary syndrome setting [34], emergency evidence supports that hs-cTnT and hs-cTnI outcome prognostic value might differ in other scenarios [35,36]. To assess if such a difference exists in this case, we will group studies by the declared essay type (i.e. hs-cTnI, hs-cTnT, hs-cTnC).

3. **Subgroup by type of association.** In order to assess whether heterogeneity may be due to the type of analysis employed to derive a dichotomous effect size, a subgroup analysis will be performed segregating studies reporting dose-response data (i.e. regressions handling hs-cTn as a continuous dependent variable) versus category/quantile-based data (i.e. contingency tables and/or regressions handling hs-cTn as a dichotomous dependent variable). This subgrouping is not applicable for the SMD model.

Furthermore, to assess whether missing adjustment for relevant confounders might be impacting the heterogeneity and pooled effect size of the mortality adjusted model: We will undertake a sensitivity analysis, meta-analyzing only effect measures of studies adjusting for (at least) the 3 predefined core confounders.

To avoid p-value overreliance, the credibility of the differential effect of statistically significant Subgroup analysis will be critically appraised with the ICEMAN questionnaire [37].

If we suspect a small study effect in any model, we will run a sensitivity analysis with a fixed effect model.

## 2.8. Publication bias

For assessment of possible small-study effect, funnel plots and Eggers test will be run for any given model that incorporates at least 10 different studies, by plotting the logarithm of the obtained odds ratios against their standard error [19,38–40]We will visually inspect the plots for signs of publication bias (namely, asymmetry

## 3. Discussion and conclusions

The prognostic significance of troponin for sepsis outcomes is not yet settled. Although previous works have consistently shown a significant association between troponin and mortality, there is a need for more research shedding light on the prognostic value of such biomarker for the most common current clinical setting in the ICU. This study will provide further evidence about the role of hs-cTn assays in predicting mortality in septic patients; potentially helping to guide further research and yielding valuable information for patient assessment.

The systematic review outline in this protocol is not exempt of possible difficulties and limitations: most notably, the failure to declare critical data, and/or the sensitivity of the troponin essays utilized by primary studies might result excluding a substantial amount of the eligible pool; another problem with troponin essays is the multiplicity of normality cutoffs, something that could end up in a synthesis of cohorts categorizing patients at different risk levels [19]; moreover, as with any meta-analysis of adjusted estimates, we expect to synthetize cohorts that considered different sets of confounders, which can introduce substantial heterogeneity in the final estimate [19].

We will evaluate the certainty of evidence for each investigated outcome with the 'Grading of Recommendations Assessment, Development and Evaluation' (GRADE) system guideline

appropriate for synthesis of prognostic factors evidence [41,42]. A "Summary of findings' table will communicate with standardized statements the certainty of this evidence

## Supporting information

**S1 Checklist. PRISMA-P (Preferred Reporting Items for Systematic review and Meta-Analysis Protocols) 2015 checklist: Recommended items to address in a systematic review protocol\*.**
(DOC)

## Author Contributions

**Conceptualization:** Santiago Ferrière-Steinert, José Ramos-Rojas, Abraham I. J. Gajardo.

**Data curation:** Santiago Ferrière-Steinert.

**Formal analysis:** Santiago Ferrière-Steinert, Joaquín Valenzuela Jiménez, Sebastián Heskia Araya, Thomas Kouyoumdjian.

**Funding acquisition:** Abraham I. J. Gajardo.

**Investigation:** Santiago Ferrière-Steinert.

**Methodology:** Santiago Ferrière-Steinert.

**Project administration:** Santiago Ferrière-Steinert.

**Resources:** Santiago Ferrière-Steinert, Abraham I. J. Gajardo.

**Supervision:** Abraham I. J. Gajardo.

**Validation:** Santiago Ferrière-Steinert, Abraham I. J. Gajardo.

**Visualization:** Santiago Ferrière-Steinert.

**Writing – original draft:** Santiago Ferrière-Steinert, Joaquín Valenzuela Jiménez, Sebastián Heskia Araya, Thomas Kouyoumdjian.

**Writing – review & editing:** Santiago Ferrière-Steinert, José Ramos-Rojas, Abraham I. J. Gajardo.

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
