## [Decision Letter · Decision Letter 0]

7 Jun 2024

PONE-D-24-11769Early high-sensitivity troponin elevation in predicting short-term mortality in sepsis: A protocol for a systematic review with meta-analysisPLOS ONE

Dear Dr. Gajardo,

Thank you for submitting your manuscript to PLOS ONE. After careful consideration, we feel that it has merit but does not fully meet PLOS ONE’s publication criteria as it currently stands. Therefore, we invite you to submit a revised version of the manuscript that addresses the points raised during the review process.

We look forward to receiving your revised manuscript.

Kind regards,

Sonu Bhaskar, MD PhD

Academic Editor

PLOS ONE

Journal Requirements:

**Additional Editor Comments:**

Thank you for submitting your work to PLOS One. Based on review of your submission and feedback from reviewers, we would like to invite you to address the points raised and revise your manuscript. We look forward to receiving your revised mansucript.

Reviewers' comments:

Reviewer's Responses to Questions

**Comments to the Author**

1. Does the manuscript provide a valid rationale for the proposed study, with clearly identified and justified research questions?

Reviewer #1: Yes

Reviewer #2: Yes

Reviewer #3: Yes

Reviewer #4: Yes

2. Is the protocol technically sound and planned in a manner that will lead to a meaningful outcome and allow testing the stated hypotheses?

Reviewer #1: Yes

Reviewer #2: Yes

Reviewer #3: Yes

Reviewer #4: Yes

3. Is the methodology feasible and described in sufficient detail to allow the work to be replicable?

Reviewer #1: Yes

Reviewer #2: Yes

Reviewer #3: Yes

Reviewer #4: Yes

4. Have the authors described where all data underlying the findings will be made available when the study is complete?

Reviewer #1: Yes

Reviewer #2: No

Reviewer #3: Yes

Reviewer #4: Yes

5. Is the manuscript presented in an intelligible fashion and written in standard English?

Reviewer #1: Yes

Reviewer #2: Yes

Reviewer #3: Yes

Reviewer #4: Yes

6. Review Comments to the Author

You may also provide optional suggestions and comments to authors that they might find helpful in planning their study.

Reviewer #1: The authors present a systematic review protocol to assess the association of early high-sensitivity serum troponin levels with short-term mortality in patients admitted to ICU because of sepsis with acute organ dysfunction. Some corrections must be made to the manuscript.

The inclusion of only studies published in English in an era of online translators and artificial intelligence is not justified in a systematic review, since texts in other languages can be translated into English and the authors' native language with good quality. The correct thing to do is not to restrict it by publication language.

Authors should consider in the methodology "The evidence for small-study effects is usually considered on a funnel plot, which shows the study estimates (x axis) against their precision (y axis). A funnel plot is usually recommended if there are 10 or more studies", following the reference https://doi.org/10.1136/bmj.k4597 .

Finally, the authors should try to anticipate in a paragraph possible limitations that could compromise the results of the future systematic review, such as observational studies with small samples, with great heterogeneity and without strict criteria for analyzing troponin levels and diagnosing sepsis.

Reviewer #2: Line 117: Studies will be first analyzed based on their title and abstract. Papers that pass this primary

scrutiny, or whenever it seems insufficient, will be subject to full text assessment by the same reviewers for final selection. (please rephrase how are you going to analyze)

Line 120: Basically, interventional and observational primary studies will be deemed eligible if they reported some form of association between troponin levels and mortality in a general clinical context (how are you going to combine the analysis for different study design).

Reviewer #3: Reviewer’s comments:

The authors are planning to perform a systematic review and meta-analysis examining the prognostic value of early high-sensitivity troponin in predicting short-term mortality in sepsis based on a detailed, comprehensive, and clear protocol that has been registered in PROSPERO (CRD42024468883).

I only have 2 minor comments on the Methods section: Selection criteria.

Page 7, Table 3.

One of the inclusion criteria only limits published articles in English language. To improve the inclusion of more potential articles, the authors may consider not to have any language restrictions.

Page 9. Table 3.

The authors plan to include published studies, conferences, and abstracts. To ensure that the quality of included studies are of higher standards, only published studies in the form of full article should be included in the systematic review and meta-analysis. Please consider excluding abstracts presented in conferences.

I have no further comments.

The end.

Reviewer #4: This study aims to verify the potential association between early serum troponin levels in high-sensitivity assays with short-term mortality risk in septic patients with acute organ dysfunction.

This protocol will produce a systematic review with meta-analysis which will try to provide further evidence about the role of high-sensitivity troponin assays in predicting mortality in septic patients, paving the way for further research in this high-risk population.

The authors have adopted solid methodology and a comprehensive statistical plan for gathering and analyzing data retrieved from available studies.

I don't have specific comments to improve the manuscript further.

My only remark relates to the sentence at line 156 : "Authors will be contacted to request critical missing or not reported data regarding associations of interest. In case of no response, the article will be excluded from this study." I wonder if the authors could find a way to include at least some data from these incomplete articles, if there were any, instead of excluding them a priori.

7. PLOS authors have the option to publish the peer review history of their article (what does this mean?). If published, this will include your full peer review and any attached files.

Reviewer #1: **Yes: **Ricardo Ney Cobucci

Reviewer #2: No

Reviewer #3: No

Reviewer #4: No

---

## [Author Response · Author response to Decision Letter 0]

23 Jul 2024

We thank the reviewers since their comments have contributed to improving the quality of our manuscript. Enclosed is our revised manuscript and a letter response point-by-point to the reviewers' comments.

Abraham IJ Gajardo

---

## [Decision Letter · Decision Letter 1]

16 Aug 2024

Early high-sensitivity troponin elevation in predicting short-term mortality in sepsis: A protocol for a systematic review with meta-analysis

PONE-D-24-11769R1

Dear Dr. Gajardo,

We’re pleased to inform you that your manuscript has been judged scientifically suitable for publication and will be formally accepted for publication once it meets all outstanding technical requirements.

Kind regards,

Sonu Bhaskar, MD PhD

Academic Editor

PLOS ONE

Additional Editor Comments (optional):

Thank you for submitting the revised version of your manuscript. I am pleased to inform you that it has been accepted for publication.

Reviewers' comments:

Reviewer's Responses to Questions

**Comments to the Author**

1. Does the manuscript provide a valid rationale for the proposed study, with clearly identified and justified research questions?

Reviewer #1: Yes

Reviewer #2: Yes

2. Is the protocol technically sound and planned in a manner that will lead to a meaningful outcome and allow testing the stated hypotheses?

Reviewer #1: Yes

Reviewer #2: Yes

3. Is the methodology feasible and described in sufficient detail to allow the work to be replicable?

Reviewer #1: Yes

Reviewer #2: Yes

4. Have the authors described where all data underlying the findings will be made available when the study is complete?

Reviewer #1: Yes

Reviewer #2: No

5. Is the manuscript presented in an intelligible fashion and written in standard English?

Reviewer #1: Yes

Reviewer #2: Yes

6. Review Comments to the Author

You may also provide optional suggestions and comments to authors that they might find helpful in planning their study.

Reviewer #1: The authors have met most of the reviewers' recommendations. The revised manuscript is ready for publication.

Reviewer #2: Dear Authors,

The background is You may change the subheading for literature search to search strategy and explain in a paragraph. You may rephrase this sentence "Subsequently, we will manually scrutinize references of included studies and previous reviews on the topic to retrieve missing pertinent papers".

7. PLOS authors have the option to publish the peer review history of their article (what does this mean?). If published, this will include your full peer review and any attached files.

Reviewer #1: **Yes: **Ricardo Ney Cobucci

Reviewer #2: No

---

## [Editor Report · Acceptance letter]

28 Aug 2024

PONE-D-24-11769R1 

PLOS ONE

Dear Dr. Gajardo, 

I'm pleased to inform you that your manuscript has been deemed suitable for publication in PLOS ONE. Congratulations! Your manuscript is now being handed over to our production team.

Kind regards, 

on behalf of

Dr. Sonu Bhaskar 

Academic Editor

PLOS ONE